# Genetic variations in *UCA1*, a lncRNA functioning as a miRNA sponge, determine endometriosis development and the potential associated infertility via regulating lipogenesis

Cherry Yin-Yi Chang[1,2☯], Li Yang[3☯], Joe Tse[4], Lun-Chien Lo[5], Chung-Chen Tseng[4], Li Sun[6,7], Ming-Tsung Lai[8], Ping-Ho Chen[9], Tritium Hwang[4], Chih-Mei Chen[10], Fuu-Jen Tsai[5,10]*, Jim Jinn-Chyuan Sheu[4,5,11,12,13]*

1 Department of Obstetrics and Gynecology, China Medical University Hospital, Taichung, Taiwan, 2 Department of Medicine, School of Medicine, China Medical University, Taichung, Taiwan, 3 Department of Gynecology, The Third Affiliated Hospital of Zhengzhou University, Zhengzhou, China, 4 Institute of Biomedical Sciences, National Sun Yatsen University, Kaohsiung, Taiwan, 5 School of Chinese Medicine, China Medical University, Taichung, Taiwan, 6 Department of Gynecological Oncology, Shandong Cancer Hospital and Institute, Shandong First Medical University and Shandong Academy of Medical Sciences, Jinan, China, 7 Department of Gynecological Oncology, Qingdao Central Hospital, The Second Affiliated Hospital of Medical College of Qingdao University, Qingdao, China, 8 Department of Pathology, Taichung Hospital, Ministry of Health and Welfare, Taichung, Taiwan, 9 School of Dentistry, Kaohsiung Medical University, Kaohsiung, Taiwan, 10 Genetics Center, China Medical University Hospital, Taichung, Taiwan, 11 Department of Biotechnology, Kaohsiung Medical University, Kaohsiung, Taiwan, 12 Institute of Biopharmaceutical Sciences, National Sun Yat-Sen University, Kaohsiung, Taiwan, 13 Institute of Precision Medicine, National Sun Yat-Sen University, Kaohsiung, Taiwan

☯ These authors contributed equally to this work.
* d0704@mail.cmuh.org.tw (FJT); sheu.jim@gmail.com (JJCS)

**Data Availability Statement:** All relevant data are within the manuscript and its Supporting Information files.

## Abstract

Endometriosis is a hormone-associated disease which has been considered as the precursor for certain types of ovarian cancer. In recent years, emerging evidence demonstrated potent roles of lncRNA in regulating cancer development. Since endometriosis shares several features with cancer, we investigated the possible involvement of cancer-related lncRNAs in endometriosis, including UCA1, GAS5 and PTENP1. By using massARRAY system, we investigated certain genetic variations in cancer-related lncRNAs that can change the thermo-stability, leading to up-regulation or down-regulation of those lncRNAs. Our data indicated three risk genetic haplotypes in *UCA1* which can stabilize the RNA structure and increase the susceptibility of endometriosis. Of note, such alterations were found to be associated with long-term pain and infertility in patients. It has been known that UCA1 can function as a ceRNA to sponge and inhibit miRNAs, resulting in loss-of-control on downstream target genes. Gene network analyses revealed fatty acid metabolism and mitochondria beta-oxidation as the major pathways associated with altered UCA1 expression in endometriosis patients. Our study thus provides evidence to highlight functional/epigenetic roles of UCA1 in endometriosis development via regulating fatty acid metabolism in women.

**Funding:** This study was supported by grants from Ministry of Science and Technology, Taiwan (110-2320-B-039-033 and 111-2320-B-039-030-MY3), Ministry of Health and Welfare, Taiwan (MHW 11017), China Medical University Hospital, Taiwan (DMR111-120), and National Sun Yat-sen University- Kaohsiung Medical University (110-P019). This study was also partially supported by grants from National Health and Family Planning Commission of Henan Province, China (2018020215 and LHGJ20200448). The funders had no role in study design, data collection and analysis, decision to publish, or preparation of the manuscript.

**Competing interests:** Competing interests: The authors declare no conflict of interest

## Introduction

Endometriosis is a condition that commonly affects 5–10% women of reproductive age globally [1]. It happens when the inner tissue layer of the uterus grows at the outside, whereas the causes are still controversial. Half of the patients suffer from severe pain during menstruation because when a woman has her period, healthy people bleed only from inside of the uterus, but endometriosis patients bleed from outside of the uterus as well. This complication causes blood contact with other organs inside the abdomen, leading to long-term inflammation, band formation and pelvic pain. For some patients (around 30–50%), they become infertile either due to distorting the fallopian tubes thus fail to pick up the egg after ovulation or due to constant inflammation that affect normal functions of the ovary, egg, fallopian tubes or uterus [2, 3]. With many biochemical changes in endometriotic lesions, the genetic/epigenetic theory was purposed in recent years to explain the hereditary aspects, the predisposition, and the endometriosis-associated changes in the endometrium, immunology, and placentation during disease development [4, 5].

Notably, endometriosis shares several phenotypes with cancer such as uncontrolled proliferation, cell migration/invasion to other surrounding organs [2]. Hormone imbalance, a proposed cause for endometriosis, can be also a risk for several types of gynecological cancer [2, 6, 7]. In fact, epidemiological and molecular investigations provided evidence to support that endometriotic lesions serve as benign precursors for certain types of gynecological cancers [8, 9]. Symptoms like recurring pelvic pain and long-term inflammation are also frequently found in patients with endometrial and uterine cancers.

In recent years, the next-generation sequencing technology has discovered thousands of long noncoding RNAs (lncRNA) in mammals. LncRNAs are non-coding RNAs with length larger than 200nt, which can undergo post-transcriptional editing just like protein-coding mRNAs, such as 5'-capping, 3'-end polyadenylation and RNA splicing [10]. Although no protein encoded, lncRNAs show dynamic impacts on gene expression through at least three well-known regulatory mechanisms: guides, dynamic scaffolds and molecular decoys [11, 12], therefore lncRNAs profiles can control the content and abundance of microRNAs (miRNAs), mRNAs, and proteins in a cell. Many studies revealed potent roles of lncRNAs in regulating carcinogenesis or tumor suppression [13]. Dysregulated expression of certain lncRNAs have been linked to the development of gynecologic cancers [14]. Some other studies also demonstrated the association of certain lncRNAs with endometriosis [15]. More interestingly, genetic alterations in those lncRNAs have been suggested as the cause to drive the pathogenesis of endometriosis [16].

UCA1 and GAS5 are lncRNAs predominantly expressed in endometrial tissues based on the HPA RNA-seq results from NCBI databank (www.ncbi.nlm.nih.gov). Previous studies found that both UCA1 and GAS5 lncRNAs play roles in the development of ovarian cancer [17]. GAS5 has been reported to function as a tumor suppressor-like gene, that can block ovary cancer progression [18–20], whereas UCA1 serves as a potential novel biomarker and therapeutic target for ovarian cancer [21, 22]. Of note, certain lncRNAs including UCA1 can interact with a variety of miRNAs which subsequently alter gene expression profiles and control cancer progression [23–25]. For examples, UCA1 can function as a competing endogenous RNA (ceRNA) to sponge tumor suppressor gene-binding miRNAs, thus promote cancer development [26–31], such as ovarian cancer [32]. On the other hand, PTENP1 is another emerging cancer-related lncRNA, which can also function as ceRNA to abolish the expression of PTEN, a well-characterized tumor suppressor, thus has been considered as a cancer-promoting gene for a variety of cancers, including endometrial cancer [9]. Although the biological consequences associated with the altered lncRNAs and miRNAs have been intensively

investigated in cancer, little is known about their possible roles in the development and progression of endometriosis. Since endometriosis has been known as the precursor of certain types of ovarian cancer, similar bio-functions of those potent lncRNAs detected in cancer may also determine the development of endometriosis.

Several lines of evidence have proven that genetic variations in miRNAs can promote endometriosis development [33]. In this study, we investigated the possible impacts of genetic variations in these ovary-related lncRNAs, *PTENP1*, *GAS5* and *UCA1*, on endometriosis. Several SNPs (single nucleotide polymorphisms) that can alter the RNA structure and thermodynamic energy were selected and genome-typed in patients and normal controls. Clinical consequences associated with the disease-related alleles were also analyzed. Furthermore, we studied the associated regulatory effects on the lncRNA-miRNA axes and discussed their involvements in the pathogenesis of endometriosis.

## Results

### Functional SNPs change thermo-energy and local structures in lncRNAs

LncRNAs have been frequently linked to hormone-sensitive cancers (S1 Table in S1 File) as well as endometriosis (S2 Table in S1 File), suggesting the involvement of those oncogenic lncRNAs in regulating cell proliferation and invasiveness. In addition, several studies also indicate that alterations in RNA structures of gene products can determine the outcomes of human diseases [34], leading to our hypothesis that alterations in RNA structure by genetic variations like SNPs may affect the progression and development of endometriosis. To prove this hypothesis, MassARRAY system was performed to detect the SNPs in oncogenic lncRNAs, including PTENP1, GAS5, and UCA1. SNPs with minor allele frequencies (MAF) larger than 10% in Chinese Han Beijing population were filtered out from NCBI databank ([www.ncbi.nih.gov/snp](www.ncbi.nih.gov/snp)). Potentially functional SNPs were further selected in which the substitutions can alter overall free energy of the lncRNAs ($\Delta\Delta G > 0.5$ or $< -0.5$ kcal/mol) and change the folding of the local RNA structures (S3 Table in S1 File). For this task, partial RNA sequences containing the SNP sites (total 121-bp, the SNP site with 60-bp upstream and 60-bp downstream sequences) were submitted to RNA stability calculation and structure prediction by using MaxExpect (rna.urmc.rochester.edu/RNAstructureWeb/Manual/MaxExpect.html). Based on those criteria and probe availability, three SNPs in *GAS5*, five SNPs in *UCA1* and four SNPs in *PTENP1* were finally picked up to further genotyping (S3 Table in S1 File). Our results show that the energy changes were valid ($\Delta\Delta G > 0.5$ or $< -0.5$ kcal/mol) in all selected SNPs, and the RNA folding got altered as compared to the wildtype sequences, including GAS5 (S1 Fig in S1 File), UCA1 (S2A Fig in S1 File) and PTENP1 (S3 Fig in S1 File). Those results confirmed the genetic variations in lncRNAs can induce changes in local RNA folding and their thermodynamics.

### Impacts of functional SNPs in lncRNAs on the susceptibility to endometriosis

Allelic and genotypic distributions of selected SNPs in this case-control study were analyzed, and our data indicated that genetic variation of A to G at rs113579010 in *UCA1* showed a tendency to increase the risk to develop endometriosis (OR = 1.718; 95% CI: 1.101–2.674) (Table 1). Dominant/recessive genotype analyses further confirmed the dominant effects of this functional SNP that associated with higher susceptibility to endometriosis (OR = 1.835; 95% CI: 1.079–3.125) (Table 2). Of note, the *p*-values were not statistically significant after Bonferroni corrections. No notable associations, allelic/genotypic distributions and dominant/

**Table 1. Genotypic and allelic distributions of five functional SNPs in *UCA1* between Taiwanese endometriosis patients and controls[1].**

| SNPs | Genotype /allele | No. (%) of patients | HWE | No. (%) of controls | HWE | *p*-value | Corrected *p*-value[2] | OR (95% CI) |
|---|---|---|---|---|---|---|---|---|
| **rs113579010** | GG | 8 | 0.854 | 3 | 0.663 | 0.520 | 1.000 | 3.322 (0.845–12.987) |
| | AG | 48 | | 35 | | | | 1.709 (0.985–2.967) |
| | AA | 61 | | 76 | | | | 1.000 Reference |
| | G | 64 (27.4%) | | 41 (13.2%) | | **0.016*** | 0.082 | **1.718 (1.101–2.674)** |
| | A | 170 (72.6%) | | 187 (86.8%) | | | | 1.000 Reference |
| rs12462414 | TT | 30 | 0.854 | 29 | 0.132 | 0.599 | 1.000 | 1.212 (0.596–2.469) |
| | GT | 61 | | 52 | | | | 1.375 (0.741–2.551) |
| | GG | 29 | | 34 | | | | 1.000 Reference |
| | T | 121 (50.4%) | | 110 (47.8%) | | 0.572 | 1.000 | 1.108 (0.772–1.592) |
| | G | 119 (49.6%) | | 120 (52.2%) | | | | 1.000 Reference |
| rs12610921 | GG | 3 | 0.000 | 3 | 0.000 | 0.787 | 1.000 | 1.333 (0.196–9.091) |
| | GA | 100 | | 91 | | | | 1.464 (0.490–4.386) |
| | AA | 6 | | 8 | | | | 1.000 Reference |
| | G | 106 (48.6%) | | 97 (47.5%) | | 0.823 | 1.000 | 1.044 (0.712–1.529) |
| | A | 112 (51.4%) | | 107 (52.5%) | | | | 1.000Reference |
| rs73003273 | AA | 8 | 0.643 | 3 | 0.616 | 0.209 | 1.000 | 2.976 (0.759–11.628) |
| | AG | 42 | | 36 | | | | 1.302 (0.750–2.257) |
| | GG | 69 | | 77 | | | | 1.000 Reference |
| | A | 58 (24.4%) | | 42 (18.1%) | | **0.097** | 0.583 | 1.458 (0.933–2.278) |
| | G | 180 (75.6%) | | 190 (81.9%) | | | | 1.000 Reference |
| rs73005445 | GG | 8 | 0.665 | 3 | 0.616 | 0.194 | 1.000 | 3.021 (0.770–11.765) |
| | AG | 42 | | 36 | | | | 1.321 (0.760–2.294) |
| | AA | 68 | | 77 | | | | 1.000 Reference |
| | G | 58 (24.6%) | | 42 (18.1%) | | **0.088** | 0.525 | 1.475 (0.943–2.304) |
| | A | 178 (75.4%) | | 190 (81.9%) | | | | 1.000 Reference |

[1] Abbreviations: SNP, single nucleotide polymorphism; HWE, Hardy–Weinberg equilibrium; OR, odds ratio; CI, confidence interval

[2] Bonferroni correction was applied to get the corrected *p*-value, which equals to *p*-value x5 in UCA1. Statistical significance: *p*-value < 0.05, *; *p*-value < 0.01, **; *p*-value < 0.001, ***.

recessive effects, were found for the selected functional SNPs in *GAS5* and *PTENP1* (S4 and S5 Tables in S1 File).

Among the SNPs tested in *UCA1*, allelic types of SNPs at rs113579010, rs73003273 and rs73005445 show lesser risks of endometriosis development (p < 0.100, Table 1), we therefore analyzed two-locus haplotypes of genetic combinations among these three SNPs. Significantly, patients with haplotypes of two allelic variants in *UCA1* showed stronger correlations with endometriosis development as compared to the controls (Table 3), suggesting more functional impacts generated by two-locus combinations than one single SNP on endometriosis development. Based on those results, we confirmed the possible involvements of three functional SNPs in *UCA1*, especially the two-locus hyplotypes among these three, in the development of endometriosis. Such genetic associations may be due to the changes in local RNA structures in UCA1 generated by genetic variations in those functional SNPs.

## Functional SNPs in UCA1 and their associations with clinical features

Based on the results by haplotype analyses, we further asked the possible relevance of those risk haplotypes with the clinical outcomes, including plasma CA125 level ($\leq$ 35.0 vs. > 35.0 U/ml), pain score (1–5 vs. 6–10), clinical stage (1 & 2 vs. 3 & 4), and reproductive activity (fertile

**Table 2. Dominant and recessive effects of five functional SNPs in *UCA1* between Taiwanese endometriosis patients and controls[1].**

| SNPs | Genotype | No. (%) of patients | No. (%) of controls | *p*-value | Corrected *p*-value[2] | OR (95%CI) |
|---|---|---|---|---|---|---|
| **rs113579010** | AG+GG | 56 (47.9%) | 38 (33.3%) | **0.025*** | 0.123 | **1.835 (1.079–3.125)** |
| | AA | 61 (52.1%) | 76 (66.7%) | | | 1.000 Reference |
| | GG | 8 (6.8%) | 3 (2.6%) | 0.134 | 0.668 | 2.717 (0.702–10.526) |
| | AA+AG | 109 (93.2%) | 111 (97.4%) | | | 1.000 Reference |
| rs12462414 | GT+TT | 91 (75.8%) | 81 (70.4%) | 0.351 | 1.000 | 1.318 (0.738–2.347) |
| | GG | 29 (24.2%) | 34 (29.6%) | | | 1.000 Reference |
| | TT | 30 (25.0%) | 29 (25.2%) | 1.000 | 1.000 | 0.988 (0.545–1.783) |
| | GG+GT | 90 (75.0%) | 86 (74.8%) | | | 1.000 Reference |
| rs12610921 | GA+GG | 103 (94.5%) | 94 (92.2%) | 0.493 | 1.000 | 1.460 (0.489–4.367) |
| | AA | 6 (5.5%) | 8 (7.8%) | | | 1.000 Reference |
| | GG | 3 (2.8%) | 3 (2.9%) | 0.627 | 1.000 | 0.934 (0.184–4.739) |
| | AA+GA | 106 (97.2%) | 99 (97.1%) | | | 1.000 Reference |
| rs73003273 | AG+AA | 50 (42.0%) | 39 (31.0%) | 0.185 | 1.000 | 1.431 (0.842–2.433) |
| | GG | 69 (58.0%) | 77 (69.0%) | | | 1.000 Reference |
| | AA | 8 (6.7%) | 3 (2.6%) | 0.134 | 0.668 | 2.717 (0.702–10.526) |
| | GG+AG | 111 (93.3%) | 113 (97.4%) | | | 1.000 Reference |
| rs73005445 | AG+GG | 50 (42.4%) | 39 (33.6%) | 0.168 | 1.000 | 1.451 (0.854–2.469) |
| | AA | 68 (57.6%) | 77 (66.4%) | | | 1.000 Reference |
| | GG | 8 (6.8%) | 3 (2.6%) | 0.129 | 0.647 | 2.740 (0.708–10.638) |
| | AA+AG | 110 (93.2%) | 113 (97.4%) | | | 1.000 Reference |

[1] Abbreviations: SNP, single nucleotide polymorphism; OR, odds ratio; CI, confidence interval

[2] Bonferroni correction was applied to get the corrected *p*-value, which equals to *p*-value x5 in UCA1. Statistical significance: *p*-value < 0.05, *; *p*-value < 0.01, **; *p*-value < 0.001, ***.

vs. infertile). Interestingly, these three potent haplotypes showed significant associations with patients who showed higher pain scores and infertility (Fig 1). Furthermore, we found that patients with those risk haplotypes show higher frequencies to harbor the lesions at cul-de-sac (Fig 2), an adhesion site that frequently associated with deeply infiltrating endometriosis and

**Table 3. Haplotype frequencies of selected SNPs in *UCA1* gene between endometriosis patient and controls[1].**

| SNPs | Genotype / allele | No. (%) in patients | No. (%) in controls | *p*-value | OR (95%CI) |
|---|---|---|---|---|---|
| rs113579010 | GA | 74 (15.7%) | 48 (10.4%) | **0.0177*** | **1.5959 (1.0821–2.3535)** |
| rs73003273 | GG | 56 (11.9%) | 36 (7.8%) | **0.0388*** | **1.5855 (1.0211–2.4618)** |
| | AA | 44 (9.3%) | 36 (7.8%) | 0.4166 | 1.2108 (0.7640–1.9189) |
| | AG | 298 (63.1%) | 340 (74.0%) | **0.0004*** | **0.6045 (0.4569–0.7996)** |
| rs113579010 | GG | 74 (15.8%) | 48 (10.4%) | **0.0154*** | **1.6121 (1.0930–2.3777)** |
| rs73005445 | GA | 52 (11.1%) | 36 (7.8%) | 0.0875 | 1.4722 (0.9425–2.2996) |
| | AG | 44 (9.4%) | 36 (7.8%) | 0.3929 | 1.2222 (0.7711–1.9372) |
| | AA | 298 (63.7%) | 340 (74.0%) | **0.0008*** | **0.6187 (0.4673–0.8192)** |
| rs73003273 | AG | 75 (15.8%) | 48 (10.3%) | **0.0139*** | **1.6209 (1.1003–2.3879)** |
| rs73005445 | AA | 43 (9.0%) | 36 (7.8%) | 0.4795 | 1.1807 (0.7435–1.8749) |
| | GG | 43 (9.0%) | 36 (7.8%) | 0.4795 | 1.1807 (0.7435–1.8749) |
| | GA | 315 (66.2%) | 344 (74.1%) | **0.0077**** | **0.6825 (0.5151–0.9044)** |

[1] Abbreviations: SNP, single nucleotide polymorphism; OR, odds ratio; CI, confidence interval

[2] Statistical significance: p-value <0.05,*; p-value <0.01,*; p-value <0.001.

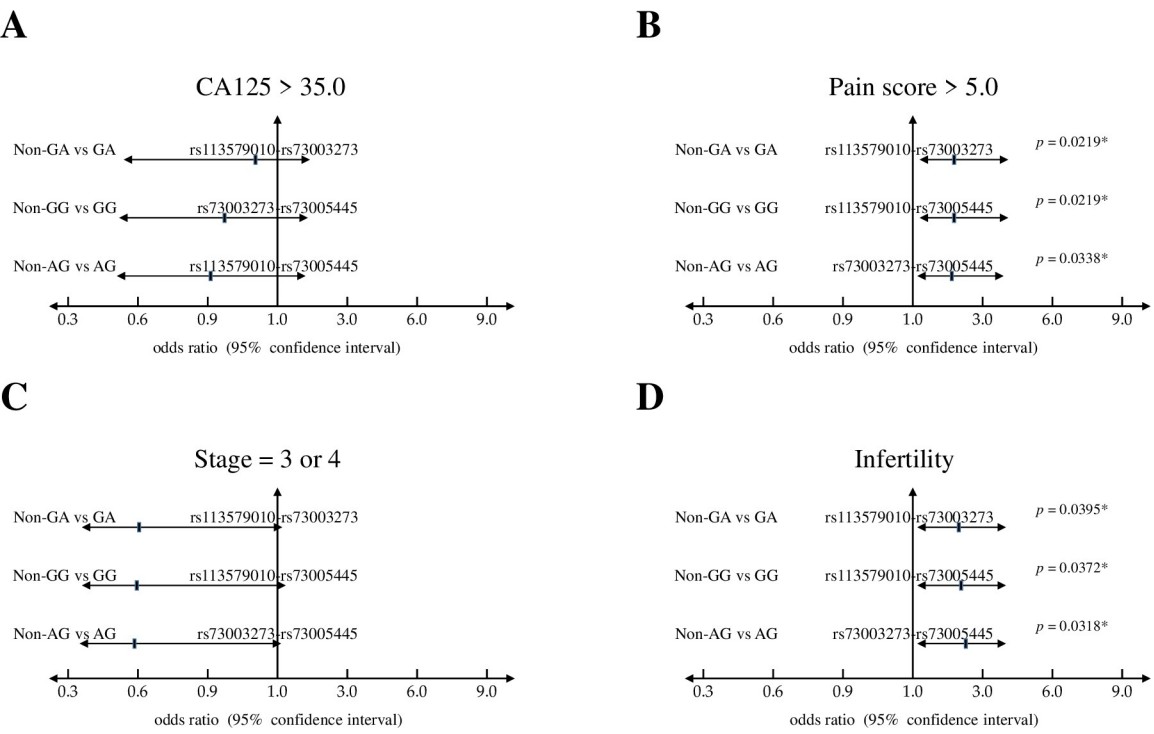

**Fig 1. Haplotype analyses of selected functional SNPs in *UCA1* and their associations with clinical features of endometriosis patients.** Odds ratios and 95% confidence intervals of three risk haplotypes based on clinical features, including (**A**) CA125 levels; (**B**) disease stage; (**C**) pain score; and (**D**) endometriosis-associated infertility.

the associated pain or infertility *[35, 36]*. Howerver, such associations did not reach statistically significance.

## Functional SNPs in UCA1 change the thermo-stability and structure of the RNA product

To learn how those novel haplotypes may affect the functions of UCA1, the full-length UCA1 variants with different haplotypes were analyzed by UNAfold web server (www.unafold.org/mfold/applications/rna-folding-form-v2.php) to estimate their thermo-stability. As shown in (Fig 3A), the optimal net energy for wild-type UCA1 was -557.2 kcal/mol; the optimal energy for the GA haplotype and the GG haplotype of rs113579010-rs73003273 were -557.4 kcal/mol (Fig 3B) and -558.0 kcal/mol (Fig 3C), respectively. For these two haplotypes of rs113579010-rs73003273, they showed similar but lower optimal net energy as compared to the wild-type. The optimal energy for the GG haplotype of rs113579010-rs73005445 was -558.3 kcal/mol (Fig 3D), which showed much lower thermo-stability than wild-type UCA1. The optimal energy for the AG haplotype of rs73003273-rs73005445 was -557.5 kcal/mol, also lower than the wild-type (Fig 3E).

In parallel with thermo-stability calculations, RNA structures of UCA1 variants with different haplotypes were also predicted. Thermo-stability analyses revealed that the GA and GG haplotypes of rs113579010-rs73003273 produce UCA1 variants with similar structures as the wild-type (Fig 4A to 4C). Notably, haplotypes associated with rs73005445, the GG haplotype of rs113579010-rs73005445 and the AG haplotype of rs73003273-73005445, can strikingly change their full-length RNA structures (Fig 4D and 4E) as compared to the wild-type. In general, our data suggest that all the risk-associated haplotypes in *UCA1* can generate more stable RNA

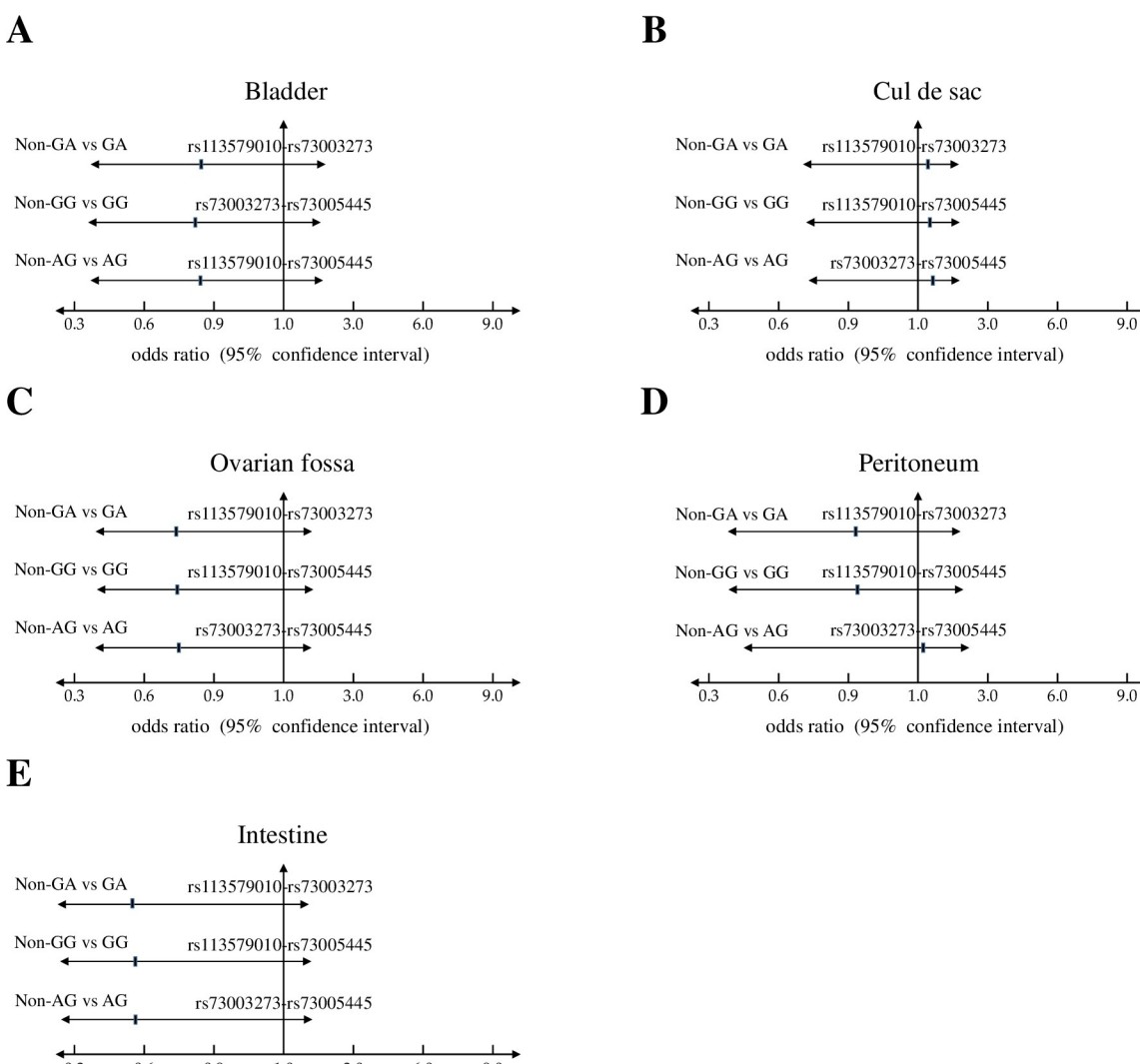

**Fig 2. Haplotype analyses of selected functional SNPs in *UCA1* and their associations with adhesion sites of endometriotic lesions.** Odds ratios and 95% confidence intervals of three risk haplotypes based on adhesion sites, including (**A**) bladder; (**B**) cul de sac; (**C**) ovarian fossa; (**D**) peritoneum; and (**D**) intestine.

products, some may even change their RNA structures, leading to higher expression UCA1 levels in endometrial tissues during the development of endometriosis.

## UCA1 functions as a miRNA sponge to disrupt lipogenesis in endometrial tissues

Previous studies discovered that elevated UCA1 can act as a miRNA sponge to change the miRNA profiles, thus control disease development and progress (S6 Table in S1 File) [37, 38]. We therefore wanted to know the possible impacts on miRNA network by elevated levels of UCA1 variants through structural stabilization. To achieve this goal, microarray data of GEO dataset (GSE120103) was used to confirm that patients with endometriosis expressed higher UCA1 levels as compared to healthy controls (p-value = 0.0083) (S4 Fig in S1 File). In addition, a miRNA network analysis associated with elevated UCA1 was done by DIANA (S7 Table in

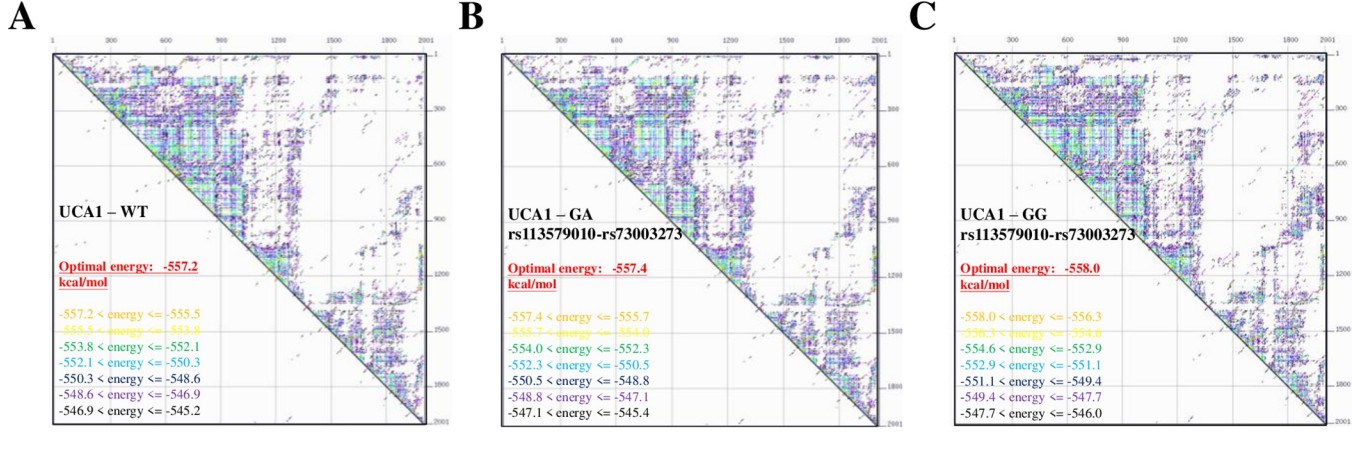

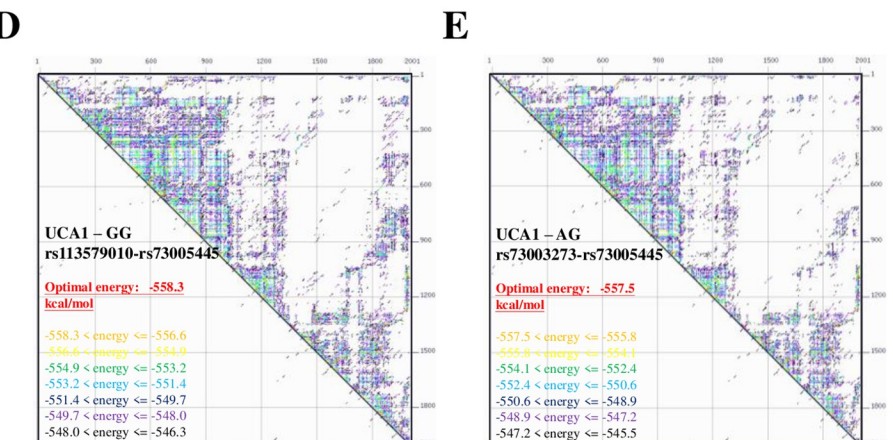

**Fig 3. Thermo-stability changes in UCA1 variants with different haplotypes.** Thermo-stability of (**A**) wild-type UCA1, (**B**) UCA1 variant with the GA haplotype of rs113579010-rs73003273, (**C**) UCA1 variant with the GG haplotype of rs113579010-rs73003273, (**D**) UCA1 variant with the GG haplotype of rs113579010-rs73005445, and (**E**) UCA1 variant with the AG haplotype of rs73003273-rs73005445. Thermo-stability of those UCA1 variants were predicted by UNAfold web server (www.unafold.org/mfold/applications/rna-folding-form-v2.php).

S1 File), and the resultant data indicated several candidate genes involved in fatty acid metabolism, biosynthesis and degradation (Fig 5A). Similar findings were also found by searching mirPath data bank (Fig 5B). To further confirm the results, we analyzed the gene expression levels of those candidate genes by using microarray data of GEO dataset (GSE120103). The most significantly relevant genes were found, including FASN, PTPLB, ACACA, ACADL, ACAT1, ACAA2, ACOX1 and HADHA, which can be elevated in endometriosis patients, especially in patients with infertility (Fig 6A). STRING protein-protein interaction analysis of those validated genes revealed their involvement in biosynthesis of unsaturated fatty acids, fatty acid metabolism/degradation/elongation, PPAR signaling pathway and fatty acid beta-oxidation pathway (Fig 6B).

## UCA1 knockdown down-regulated enhanced lipogenesis in endometrial cells

To further validate the possible functions of UCA1 in fatty acid metabolism in endometrial cells, UCA1 knockdown was performed by co-transfecting cells with pDECKO_UCA1 and lentiCas9 vectors. The selected cell lines included endometrial cancer cells (RL-95-2, HEC1A),

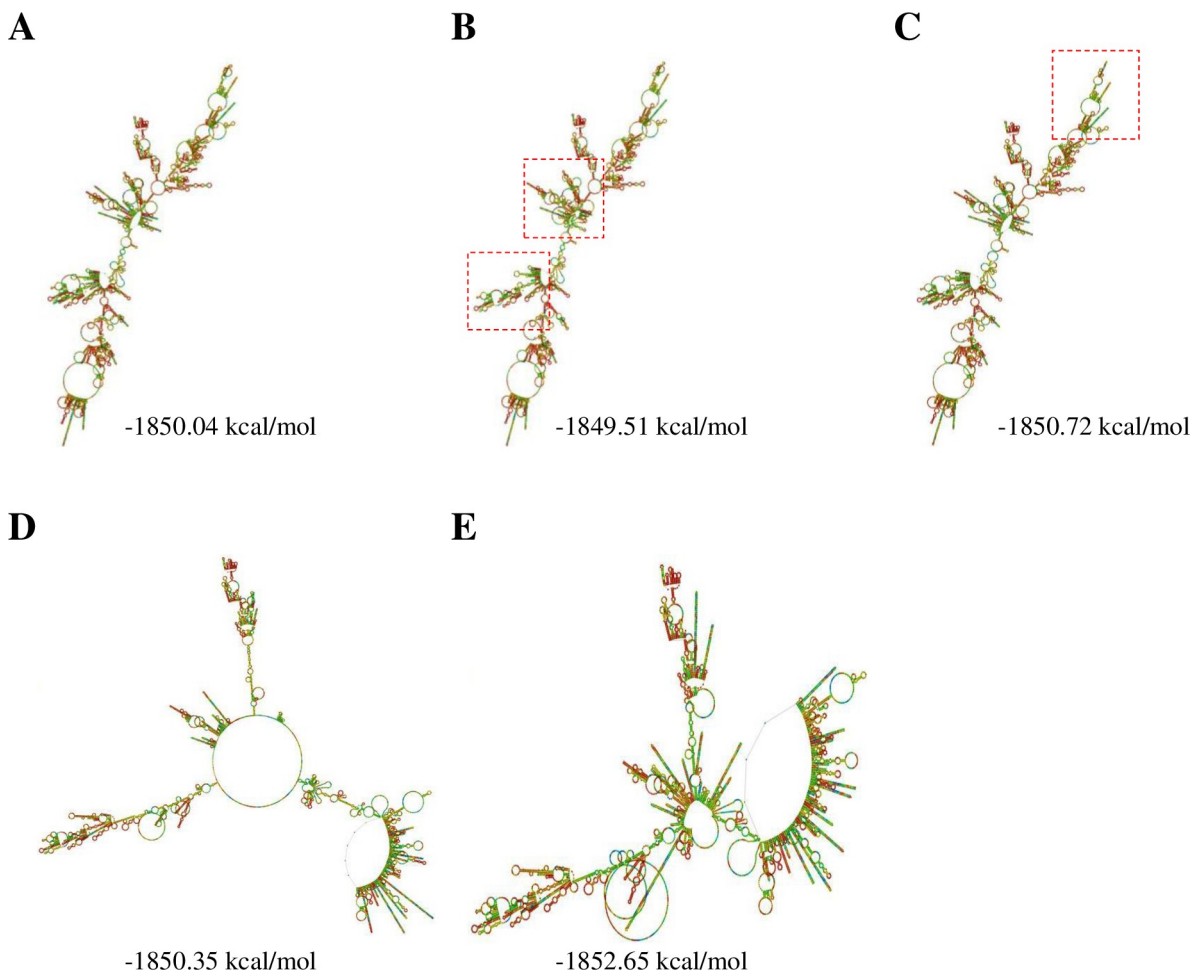

**Fig 4. Changes in RNA structures of UCA1 variants with different haplotypes.** The predicted RNA structures of (**A**) wild-type UCA1, (**B**) UCA1 variant with the GA haplotype of rs113579010-rs73003273, (**C**) UCA1 variant with the GG haplotype of rs113579010-rs73003273, (**D**) UCA1 variant with the GG haplotype of rs113579010-rs73005445, and (**E**) UCA1 variant with the AG haplotype of rs73003273-rs73005445. Full-length RNA strucures of those UCA1 variants were predicted by RNAfold web server (rna.tbi. univie.ac.at/cgi-bin/RNAWebSuite/RNAfold.cgi). Dash line boxes indicate the altered local structures in the RNA transcripts.

clear cell type ovarian cancer cells (endometriosis-origin; ES-2, TOV-21G) and human uterus fibroblast (HUF, as the normal cell control). After drug selection and cell enrichment, RNA samples were extracted from the transfected cells and subjected to RT-qPCR analyses. Our qPCR results indicate that most of the transfected cells showed reduced UCA1 levels (Fig 7A). In particular, UCA1 was downregulated massively in ES-2 cells. Consistent with our previous findings, ACADL, ACAT1, and ACAA2 genes, the major regulators in fatty acid degradation and beta-oxidation pathways, were significantly down-regulated (Fig 7B). Lipid droplet staining by LipidTOX-Red further confirmed accumulated lipids in treated cells after UCA1 knockdown (Fig 7C). With those results described above, we can conclude that elevated UCA1 in endometriosis patients may act as a miRNA sponge to alter miRNA profiles which favor gene expression of key regulators in fatty acid metabolism.

## Discussion

In this study, we discovered the involvement of novel haplotypes of *UCA1* gene in endometriosis development and the associated clinical outcomes, including long-term pain and

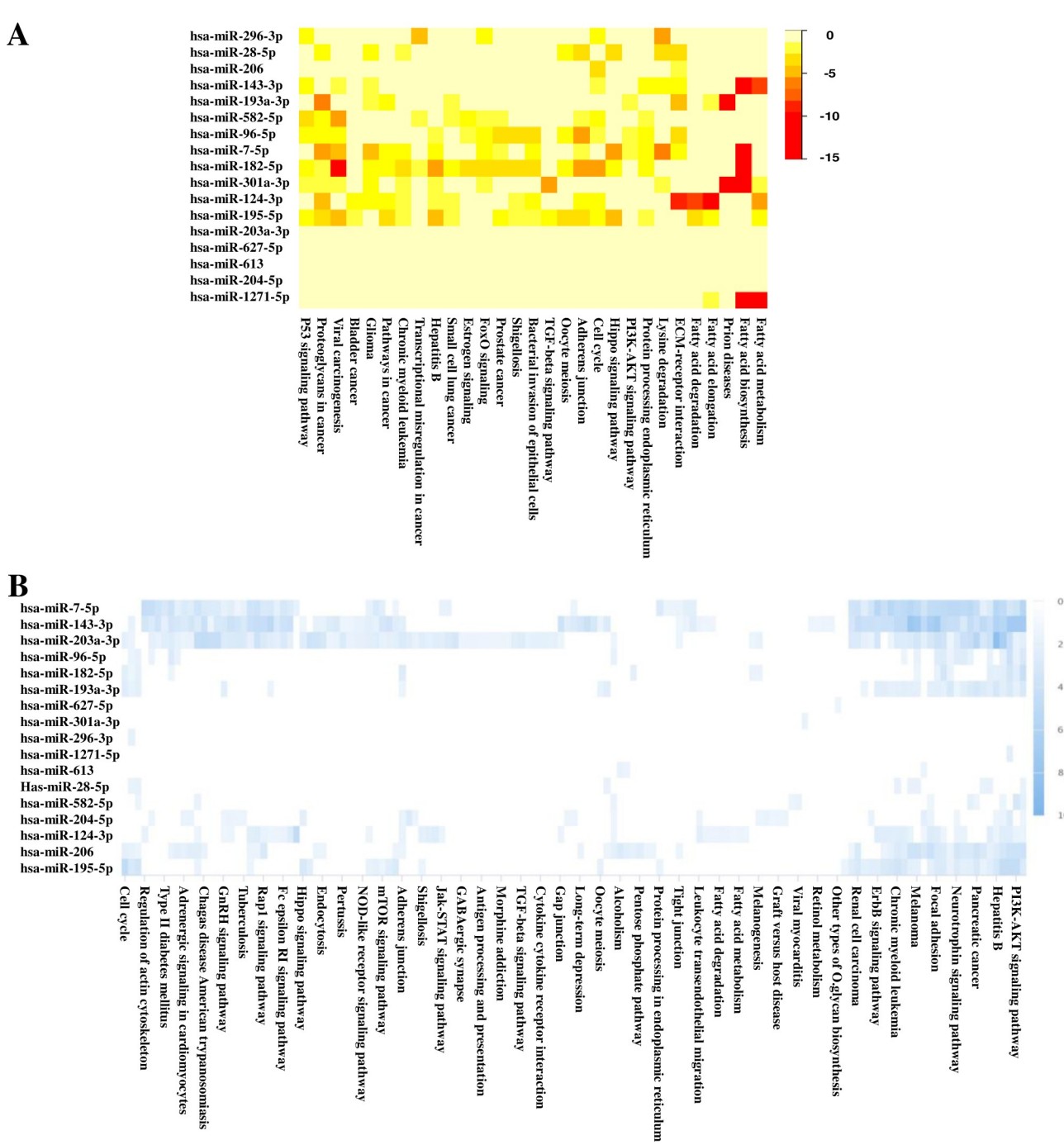

**Fig 5. UCA1 functions as a miRNA sponge to remodel the miRNA networks.** The miRNA networks associated with UCA1 elevation were predicted by (**A**) DIANA Tools mirPath and (**B**) miRPathDB v2.0. Heat map analyses indicate the statistic significances (-log p-values) of major down-stream pathways regulated by the altered miRNA networks.

infertility (Tables 1 to 3 and Fig 1). Those risk haplotypes are composed by the genetic combinations among three functional SNPs at rs113579010 (allele G), rs73003273 (allele A) and rs73005445 (allele G). Most of patients who carry those risk haplotypes show the tendency to harbor cul-de-sac infiltration (Fig 2), which can partially explain their associations with long-term pain and infertility. Since those functional SNPs were selected due to their ability to alter

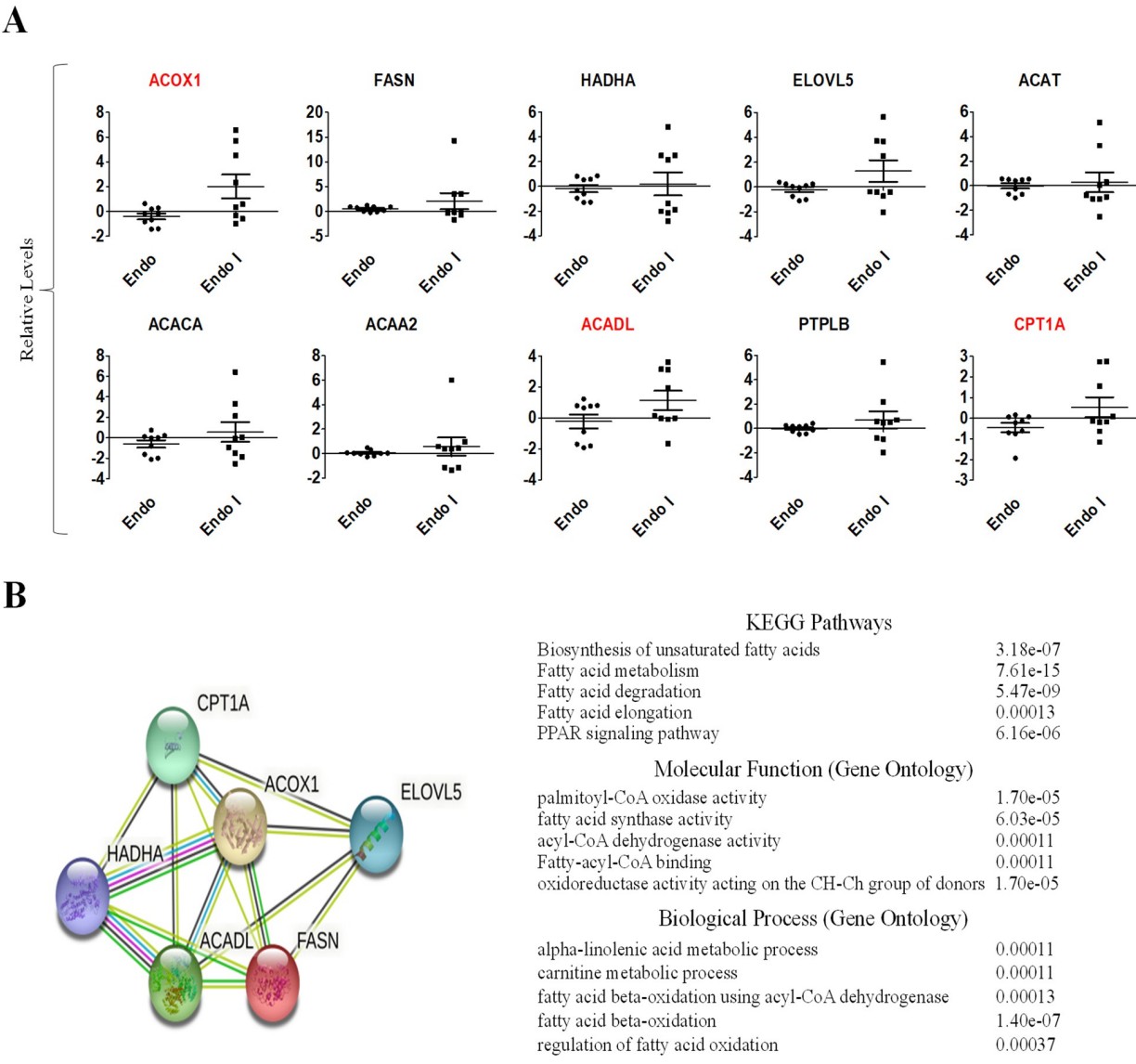

**Fig 6. Expression levels of key effector genes of UCA1 as being a miRNA sponge in endometrial patients with/without infertility.** (A) Expression levels of UCA1 down-stream genes involved in fatty acid metabolism pathways were analyzed by using microarray data of GEO databank (GSE120103). Endo: patients without infertility; Endo-I: patients with infertility. (B) Key pathways associated with altered down-stream effectors were analyzed by STRING protein-protein interactome databank.

thermo-stability, we further investigated the changes in optimal net energy and RNA structures of those UCA1 variants by RNA folding prediction. Our data confirmed that most of the UCA1 variants with those risk haplotypes showed reduced free-energy/enhanced stability (Fig 3), which can subsequently alter their full-length RNA structures (Fig 4), suggesting elevated UCA1 in endometriotic lesions. Considering the potent roles of UCA1 as a miRNA sponge, we further defined several pathways involved in fatty acid metabolism as the key down-stream signaling networks induced by altered miRNA profiles after UCA1 up-regulation (S5 Fig and S6 Table in S1 File). Transcriptome analysis by using dataset from GSE120103 confirmed increased levels of UCA1 in patients with endometriosis (S4 Fig in S1 File). Interestingly, several key regulatory genes involved in fatty acid metabolism were also found up-regulated in

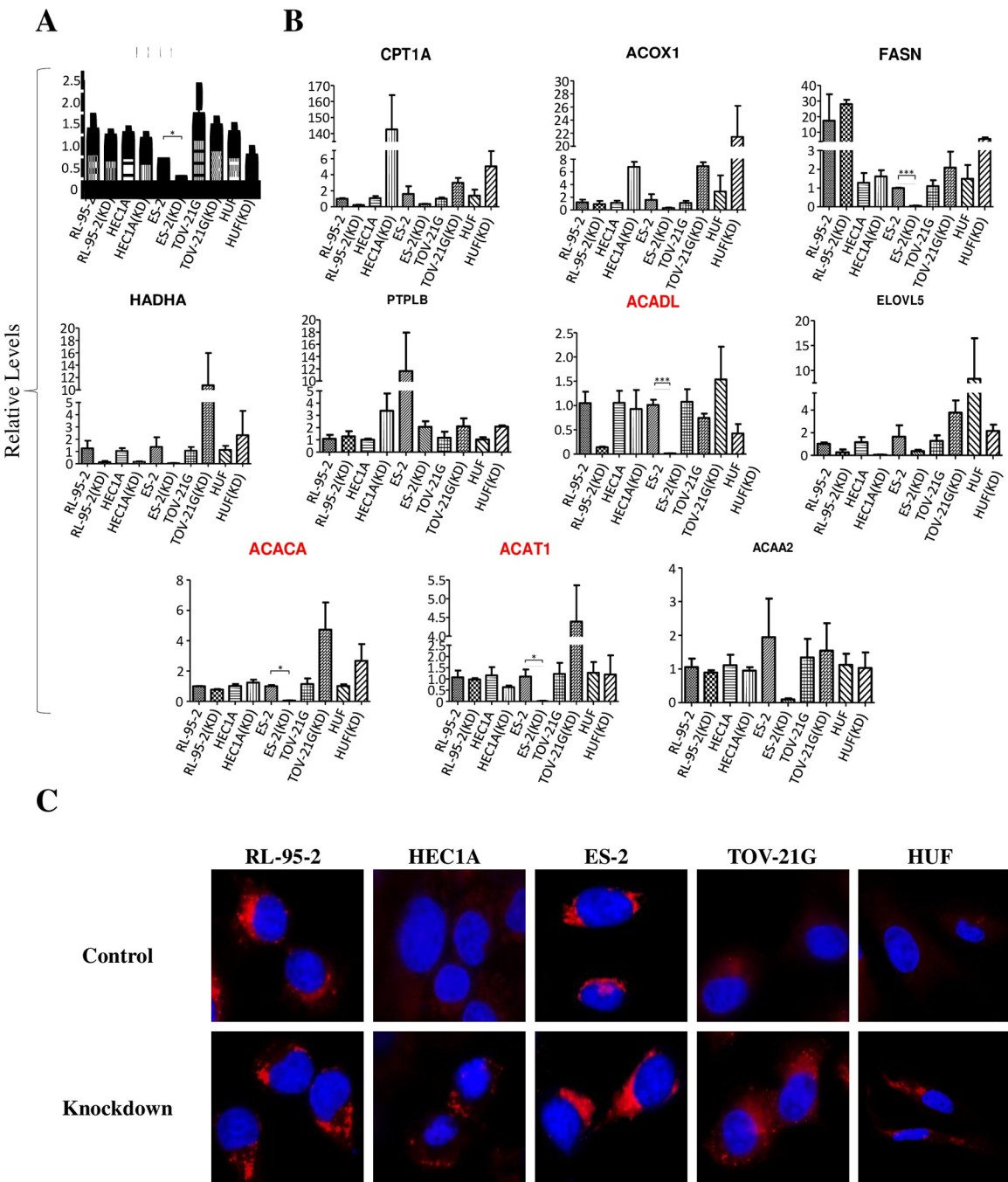

**Fig 7. Functional impacts of UCA1 knockdown on fatty acid metabolism in endometrial cells.** (**A**) UCA1 knockdown in different endometrial cell lines (RL-95-2, HEC1A, ES-2, TOV-21G, and HUF). (**B**) Relative gene expression levels of key genes in fatty acid metabolism (CPT1A, ACOX1, FASN, HADHA, ACADL, ELOVL5, ACACA, ACAT1, PTPLB, ACAA2) were detected by RT-qPCR in UCA1-knockdown and control cells. (**C**) Lipid droplet staining in treated cells was performed by using LipidTOX-Red.

patients, especially for the ones with infertility (Fig 6). Functional knockdown of UCA1 in endometrial cells can significantly suppressed expression levels of key genes in lipogenesis (ACADL, ACAT1, and ACAA2 genes), resulting in accumulation of lipid droplets in the treated cells (Fig 7). Our study therefore reveals a novel role of UCA1 in endometriosis

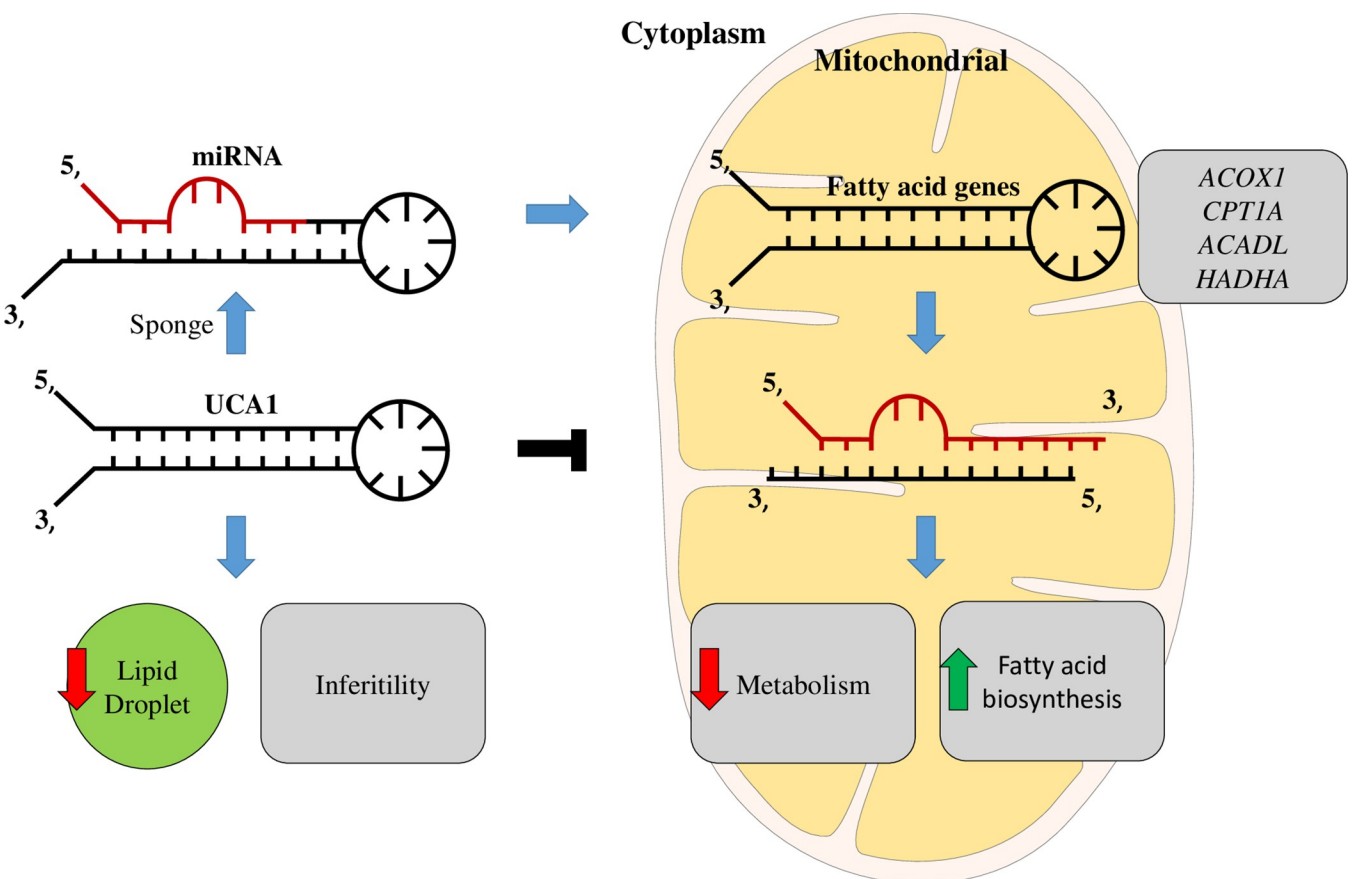

**Fig 8. Proposed pathogenesis model in the development of endometriosis and the associated infertility based on the findings of this study.** Elevated UCA1 in endometriosis patients via RNA stabilization by functional SNPs can disrupt lipogenesis regulation by sponging key miRNAs with target genes involved in lipid metabolism. Such epigenetic regulation can further lead to reduced biosynthesis of triacyglycerol for oocyte future maturation and increased lipotoxicity, resulting in infertility.

development and the associated infertility through regulating genes involved in fatty acid metabolism and lipogenesis (Fig 8).

UCA1 overexpression has been reported in different types of cancer, such as breast cancer, ovarian cancer, bladder cancer, and hepatocellular carcinoma [26, 27, 39, 40]. In addition to its potent roles in tumorigenesis, UCA1 overexpression was also frequently linked with advanced cancerous behaviors, including metastasis [28, 30], drug resistance [26, 40] and anti-immune surveillance [41]. As endometriosis shares several similar phenotypes with cancer including uncontrolled cell proliferation and long-term inflammation, this study reported a novel mechanism in endometriosis development by upregulating UCA1 through stabilizing RNA structures of the risk haplotype variants. Based on UCA1 sponging miRNA profile analysis (Fig 4), cell cycle up-regulation is one of downstream pathways associated with UCA1 elevation, suggesting the advantages for lesion growth. In addition, patients with those risk haplotypes also showed higher pain scores, which may also indicate long-term inflammation associated with UCA1 elevation, probably due to consistent lesion growth.

Another interesting finding in this study is the involvement of fatty acid metabolism associated with increased UCA1 levels (Figs 5 to 7). Although a previous study has revealed a possible connection between fatty acids (e.g. palmitate acid) and UCA1 [42], no solid evidence was provided to confirm the direct relationship between UCA1 and fatty acid metabolism pathway.

Through miRNA network analysis, we have found that UCA1 can sponges certain miRNAs that target genes involved in fatty acid metabolism, including FADS2, ELOVL5, ACOX1, CPT1A, ACADL and HADHA (Fig 5). Among those genes, ACOX1, CPT1A, ACADL and HADHA which participate in fatty acid beta-oxidation can be further validated by clinical samples (Fig 6) and cell-based study (Fig 7). This new finding provides a new insight on the molecular link between UCA1 and fatty acid metabolism. Those findings may explain and support the involvement of UCA1 in palmitate acid-induced cancer aggressiveness [42].

In our study, we also found that endometriosis patients with higher UCA1 were more susceptible to being infertile, part of the reason may be due to unhealthy conditions of the oocyte. Oocyte maturation is an essential process to make women fertile, but this process could be easily affected by the energy statuses of cumulus cells surrounding the oocyte [43]. Cumulus cells are supportive cells that can produce lipid droplets to protect and supply energy to oocytes [44]. One possible scenario associated with increased infertility in patients with risk haplotypes of *UCA1* would be that elevated UCA1 can enhance beta-oxidation in cumulus cells, resulting in the shortage of lipid droplet storage. Such energy-exhausted condition may limit the cells to supply sufficient energy to oocytes during oocyte maturation, thus reduce the quality of oocytes in patients (Fig 8). It has been known that women with low BMI, especially Asian women, are at higher risk to become infertile than those with normal or higher BMI [45]. Clinical observations also revealed that lower BMI is a risk factor for endometriosis development and such association is higher in patients with infertility [46–48].

Although our data indicated the involvement of UCA1 genetic variants in endometriosis development and the associated infertility, limitations in this study should be addressed here. Firstly, this study should be considered as a pilot study due to a relatively small sample size. A validation study by using another cohort with a bigger sample size can further elucidate the potent impacts of those functional SNPs and risk haplotypes on this disease. Secondly, we currently only confirmed the functional links between UCA1 levels and lipid droplet storage by *in vitro* cell line-based studies. *In vivo* study by using *UCA1* gene knock-out mice may be able to further confirm how UCA1 influences fatty acid metabolism in the reproductive system and infertility development.

## Materials and methods

### Clinical samples and patient information

Blood samples were collected from study subjects who were recruited at China Medical University Hospital, Taiwan. Both patients and healthy women were confirmed by laparotomy or laparoscopy examines in this prospective study. This study was approved by the Institutional Review Board (IRB) at the CMUH (CMUH106-REC1-138) with informed consent from each participant. For the control group, we selected a healthy cohort with matched age profiles as the patient group. Clinical information of the study subjects was collected from clinical notes. Endometriosis stages were classified according to the guidelines of the American Society of Reproductive Medicine (ASRM) as stage 1, 2, 3, and 4, respectively [49].

### Genome typing by MassARRAY

DNA samples were extracted from blood samples of study subjects by using Genomic DNA isolation kit (Qiagen, Valencia, CA, USA). MassARRAY system (S5A Fig in S1 File) were used to verify genetic variations of the selected functional SNPs in different lncRNA genes. In this MassARRAY system (Agena Bioscience, San Diego, CA, USA), PCR primers were designed (as shown in S3 Table in S1 File) to amplify the target regions followed by shrimp alkaline phosphatase (SAP) treatment to dephosphorylate excess nucleotides. The treated samples were

further subjected to iPLEX extension reaction with extension primers that can anneal into the specified SNP sites of the amplified DNA fragments and the reactions can be terminated by terminator nucleotides. The final extension products (analytes) were desalted and transferred onto a SpectroCHIP® Array with an automated nano-dispenser. MALDI-TOF-MS was used to perform matrix assisted laser desorption ionization that can differentiate the sizes of SNP variations (S5B Fig in S1 File).

## RNA structure and thermo-stability prediction

To study the structure and thermo-stability of the lncRNAs with genetic substitutions at the defined SNP sites, local sequences (total 121bp, the SNP site with the sequences of upstream 60 bp and downstream 60bp) were submitted to MaxExpect platform (rna.urmc.rochester.edu/RNAstructureWeb/Servers/Predict1/Predict1.html) to analyze the RNA stability and structure. As for the UCA1 variants with different haplotypes, thermo stability of the selected region (total 2001bp, the rs113579010 SNP site with the sequences of upstream 1000 bp and downstream 1000 bp which can also cover the other two SNP sites) were predicted by UNAfold platform (www.unafold.org/mfold/applications/rna-folding-form-v2.php). The full structures of UCA1 variants were predicted by RNAfold platform (rna.tbi.univie.ac.at/cgi-bin/RNAWebSuite/RNAfold.cgi). The RNA stability was estimated for alterations in the overall free energy change ($\Delta\Delta G$).

## Statistical analysis

Distributions of allelic and genotypic frequencies of selected SNPs were compared between healthy controls and endometriosis patients by 2X2 chi-square analysis (vassarstats.net/odds2x2.html). The total numbers and frequencies of allelic types, genotypes and haplotypes were compared between controls and patients and presented in different contingency tables with 95% confidence intervals (95% CIs) and odds ratio (ORs). The dominant alleles were utilized as the references.

## Pathway analysis

DIANA Tools mirPath (snf-515788.vm.okeanos.grnet.gr/index.php?r = mirpath) was utilized to analyze multi miRNA networks and the altered Kyoto Encyclopedia of Genes and Genomes (KEGG) pathways according to expression levels of downstream target genes. On the other hand, miRPathDB v2.0 heatmap calculator (mpd.bioinf.uni-sb.de/heatmap_calculator.html?organism = hsa) was used to calculate miRNA network. The $p$-values of major altered pathways were calculated by Fisher's exact test. Differences of the hypergeometric distributions were considered statistically significant when $p$-values $< 0.050$. STRING database (string-db.org/cgi/input.pl?sessionId=DdLKh4m0pH4j&input_page_show_search=on) was used to predict protein-protein interactions between different effector genes.

## Cell culture & UCA1 knockdown

Ovarian clear cancer cells (TOV-21G and ES-2) and endometrial cells (RL95-2 and HEC1A) were purchased from Bioresource Collection and Research center (BRCR), Taiwan. Human uterine fibroblast (HUF) cell line was purchased from ScienCell Research Laboratories (SC-7040; Carlsbad, CA, USA). The cells were maintained in DMEM medium containing 10% FBS and 1% penicillin/streptomycin (Gibco/Thermo Fisher Scientific, Waltham, MA, USA). Vector pDECKO_UCA1 (Addgene, Watertown, MA) which expresses two gRNAs targeting *UCA1* promoter was used to knockdown *UCA1* by co-transfection with lentiCas9-Blast

(Addgene, Watertown, MA) into cells. The effected cells were further enriched by 5 μg/ml puromycin in the culture medium.

## Supporting information

**S1 File.**
(RAR)

## Acknowledgments

The authors thank Dr. Kuan-Hao Tsui (Kaohsiung Veterans General Hospital, Taiwan) for his critical comments on this study. The authors also thank Dr. Senthilkumar Ravichandran (National Sun Yatsen University, Taiwan) for his technical assistance.

## Author Contributions

**Conceptualization:** Cherry Yin-Yi Chang, Li Yang, Jim Jinn-Chyuan Sheu.

**Data curation:** Cherry Yin-Yi Chang, Jim Jinn-Chyuan Sheu.

**Formal analysis:** Li Yang, Lun-Chien Lo, Li Sun, Ping-Ho Chen.

**Funding acquisition:** Cherry Yin-Yi Chang, Li Yang, Ping-Ho Chen, Jim Jinn-Chyuan Sheu.

**Investigation:** Joe Tse, Chung-Chen Tseng, Tritium Hwang.

**Methodology:** Li Yang, Joe Tse, Chung-Chen Tseng, Chih-Mei Chen.

**Project administration:** Tritium Hwang, Chih-Mei Chen.

**Resources:** Cherry Yin-Yi Chang, Jim Jinn-Chyuan Sheu.

**Software:** Joe Tse, Chung-Chen Tseng.

**Supervision:** Fuu-Jen Tsai, Jim Jinn-Chyuan Sheu.

**Validation:** Cherry Yin-Yi Chang, Li Yang, Lun-Chien Lo, Ming-Tsung Lai, Ping-Ho Chen, Fuu-Jen Tsai, Jim Jinn-Chyuan Sheu.

**Visualization:** Joe Tse, Jim Jinn-Chyuan Sheu.

**Writing – original draft:** Cherry Yin-Yi Chang, Joe Tse, Lun-Chien Lo.

**Writing – review & editing:** Fuu-Jen Tsai, Jim Jinn-Chyuan Sheu.

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
