## [Decision Letter · Decision Letter 0]

11 May 2022

PONE-D-22-10884Genetic variations in UCA1, a lncRNA functioning as a miRNA sponge, determine endometriosis development and the associated infertility via regulating lipogenesisPLOS ONE

Dear Dr. Sheu,

Thank you for submitting your manuscript to PLOS ONE. After careful consideration, we feel that it has merit but does not fully meet PLOS ONE’s publication criteria as it currently stands. Therefore, we invite you to submit a revised version of the manuscript that addresses the points raised during the review process.

As you can see from comments appended below, while the reviewers find your work of interest, they have raised points that need to be addressed particularly concerning clinical samples and patient information.

We look forward to receiving your revised manuscript.

Kind regards,

Giovanni Nassa, PhD

Academic Editor

PLOS ONE

Journal Requirements:

Reviewers' comments:

Reviewer's Responses to Questions

**Comments to the Author**

1. Is the manuscript technically sound, and do the data support the conclusions?

Reviewer #1: Partly

Reviewer #2: Yes

2. Has the statistical analysis been performed appropriately and rigorously? 

Reviewer #1: I Don't Know

Reviewer #2: Yes

3. Have the authors made all data underlying the findings in their manuscript fully available?

Reviewer #1: Yes

Reviewer #2: Yes

4. Is the manuscript presented in an intelligible fashion and written in standard English?

Reviewer #1: Yes

Reviewer #2: Yes

5. Review Comments to the Author

Reviewer #1: The authors present a study about the role of genetic variations in UCA1, a lncRNA functioning as a miRNA sponge in endometriosis development. In common with a lot of other lncRNAs, UCA1 can regulate the transcription of genes via epigenetic modification. This is a topic of great interest , considering the genetic–epigenetic theory of the etiopathogenesis of endometriosis that is emerging in recent years.

However, it is clear that the pathways involved in endometriosis are complicated, and the molecular mechanisms that underlie the process are largely elusive.

The results reported want to add more information about miRNA sponge in endometriosis development

Different concerns were raised that preclude acceptance in its current form.

These are major limitations which should be addressed.

TITLE

1) I suggest to evaluate to change the title in : “ Genetic variations in UCA1, a lncRNA functioning as a miRNA sponge, determine endometriosis development and the potential associated infertility via regulating lipogenesis”

INTRODUCRION

2) The introduction should focus also on the genetic–epigenetic theory of the etiopathogenesis of endometriosis that is emerging in recent years. The authors could identify other literature on the topic and explain how the study relates to this previously published research on endometriosis.

3) Futhermore the introduction lacks some bibliographical references:

- “For some patients (around 30-50%), they become infertile either due to distorting the fallopian tubes thus fail to pick up the egg after ovulation or due to constant inflammation that affect normal functions of the ovary, egg, fallopian tubes or uterus”

- “ Hormone imbalance, a proposed cause for endometriosis, can be also a risk for several types of gynecological cancer”

RESULTS

4) In the section about “ Functional SNPs in UCA1 and their associations with clinical features”

The authors correlate the “a potent role of UCA1 in endometriosis development”. From the literature we now that there is not clear correlation with the extension of the disease and the pain score, so the results from this section that the UCA1 can have a role in endometriosis development according to the pain could be not completely correct.

5) Futherome could be interesting to analyze the correlation between the localization of endometriosis ( ovarian, peritoneal or deep endometriosis ) and the Functional SNPs in UCA1.

DISCUSSION

6) The discussion should be focus more on the potential roles of UCA1 in endometriosis , because the results are not so consistent to support a certain involvement of novel haplotypes of UCA1 gene in endometriosis development and the associated clinical outcomes. Furthermore the result on association with including long-term pain and infertility are still not so consistent , but with a potential clinical relevance.

METHODS Clinical samples and patient information

7) The authors should tell us how the control group were confirmed to be healthy . Minimal or mild endometriosis is difficult to detect with imaging and without a laparoscopy the presence of endometriosis can’t be confirmed

8) better specify the type of study ( prospective or retrospective study )

Reviewer #2: The authors provided a clear and well-written paper. The results and the statistical data provided are scientifically sound and the limitations of their findings were clearly assessed in the discussion section. For this reason, I cannot find major revisions.

However, some minor revisions should be assessed:

1) Page 3 - "Those results confirmed the genetic vibrations in"  variations?

2) Page 5 - "potent" is written 3 times in 3 consecutive sentences. Please, use a synonym

3) Page 5 - "Thermo-stability data, the haplotype with rs73005445 can strikingly change their full-length RNA structures (Fig. 3D and 3E)" - Is something missing?

4) Page 5 - Figure 3B is not cited in the text.

5) Page 7 - Figure 4 labels are very small. It should be better to increase a bit their size.

6) Page 10 - "Regulation of fatty acid... "  I appreciate a good recap in biochemistry, but I don't understand the point of doing this in the discussion section. Is there some final sentence missing? What's the point of explaining all this and not connecting it with the text? I suggest adding a sentence or rephrasing the entire part.

7) Page 11 - "Blood samples were collected patients"  "Blood samples were collected from patients"

8) Page 12 - "were further enriched by 5"  A square appears after the 5. Some symbol is not recognized, please correct it.

9) Supplementary - I found many small typos in the pathways found: "Giloma" -> "Glioma"?, "Lysince degradation" -> "Lysine degradation"? etc... Please, check carefully the entire document and correct them all.

6. PLOS authors have the option to publish the peer review history of their article (what does this mean?). If published, this will include your full peer review and any attached files.

Reviewer #1: No

Reviewer #2: No

---

## [Author Response · Author response to Decision Letter 0]

23 Jun 2022

Comments from the Reviewer #1

The authors present a study about the role of genetic variations in UCA1, a lncRNA functioning as a miRNA sponge in endometriosis development. In common with a lot of other lncRNAs, UCA1 can regulate the transcription of genes via epigenetic modification. This is a topic of great interest, considering the genetic–epigenetic theory of the etiopathogenesis of endometriosis that is emerging in recent years.

However, it is clear that the pathways involved in endometriosis are complicated, and the molecular mechanisms that underlie the process are largely elusive.

The results reported want to add more information about miRNA sponge in endometriosis development

Different concerns were raised that preclude acceptance in its current form.

These are major limitations which should be addressed.

1. I suggest to evaluate to change the title in: “Genetic variations in UCA1, a lncRNA functioning as a miRNA sponge, determine endometriosis development and the potential associated infertility via regulating lipogenesis”

Response: Thank you for this constructive suggestion. Due to limited sample size, we agree that our data in this study may not be able to fully support our conclusion. We have changed the title according to the suggestion.

2. The introduction should focus also on the genetic–epigenetic theory of the etiopathogenesis of endometriosis that is emerging in recent years. The authors could identify other literature on the topic and explain how the study relates to this previously published research on endometriosis.

Response: Thank you for the invaluable suggestion. We have updated some proposed theories of endometriosis etiology into the introduction. In the first paragraph of the introduction, we inserted ‘‘With many biochemical changes in the endometriotic lesions, the genetic/epigenetic theory was purposed in recent years to explain the hereditary aspects, the predisposition, and the endometriosis-associated changes in the endometrium, immunology, and placentation during endometriosis development’’. Two references (Refs. 4, 5) were added. The references were reordered accordingly.

3. Furthermore the introduction lacks some bibliographical references:

- “For some patients (around 30-50%), they become infertile either due to distorting the fallopian tubes thus fail to pick up the egg after ovulation or due to constant inflammation that affect normal functions of the ovary, egg, fallopian tubes or uterus”

- “Hormone imbalance, a proposed cause for endometriosis, can be also a risk for several types of gynecological cancer”

Response: Thank you for those helpful suggestion. We have inserted bibliographical references for endometriosis associated infertility (Refs. 2 and 3) and hormone imbalance induced gynecological cancers (Refs. 2, 6, 7).

4. In the section about “Functional SNPs in UCA1 and their associations with clinical features”

The authors correlate the “a potent role of UCA1 in endometriosis development”. From the literature we know that there is not clear correlation with the extension of the disease and the pain score, so the results from this section that the UCA1 can have a role in endometriosis development according to the pain could be not completely correct.

Response: Thank you for the insightful comment. Due to relatively small sample size, we agree that our clinical association data may not be completely correct. We have inserted the limitation of this study in the last paragraph in the discussion in page 13. The last conclusive sentence in this section was also removed.

5. Furthermore, it could be interesting to analyze the correlation between the localization of endometriosis (ovarian, peritoneal or deep endometriosis) and the Functional SNPs in UCA1.

Response: Thank you for the insightful comment. We performed association study to know the correlation of lesion adhesion site with functional SNPs in UCA1. We found that patients with risk haplotypes (G-A of rs113579010-rs73003273, G-G of rs113579010-rs73005445, and A-G of rs73003273-rs73005445) show higher frequencies to harbor the lesions at cul-de-sac, an adhesion site that frequently associated with deeply infiltrating endometriosis and the associated pain or infertility (35, 36). However, such associations did not reach statistically significance. A new figure (Fig. 2 was added), thus the figures were reordered accordingly). Two additional references (Refs. 35, 36) were added to support the possible link of cul-de-sac infiltration with pain and infertility. We do not have clear clinical data about the DIE, thus failed to perform association study of risk haplotypes with DIE. 

6. The discussion should be focus more on the potential roles of UCA1 in endometriosis, because the results are not so consistent to support a certain involvement of novel haplotypes of UCA1 gene in endometriosis development and the associated clinical outcomes. Furthermore, the result on association with including long-term pain and infertility are still not so consistent, but with a potential clinical relevance.

Response: Thank you for the insightful comment. We removed some biochemistry parts on lipogenesis and focused more on the epigenetic roles of UCA1 variants in endometriosis development as shown in page 12-13. As what we have learnt from the new Fig.2, patients with risk haplotypes may easily get cul-de-sac infiltration, even though the data did not reach statistically significance. Such finding may partially explain why patients with risk haplotypes suffer from long-term pain and infertility. 

7. The authors should tell us how the control group were confirmed to be healthy. Minimal or mild endometriosis is difficult to detect with imaging and without a laparoscopy the presence of endometriosis can’t be confirmed

Response: Thank you for the question. Yes, both patients and healthy women were confirmed by laparotomy or laparoscopy examines. We also rearranged the text in the “Clinical samples and patient information” section of M&M part in page 13-14 to make it clear.

8. better specify the type of study (prospective or retrospective study)

Response: Thank you for the suggestion. It was a prospective study approved by the Institutional Review Board (IRB) at the CMUH (CMUH106-REC1-138) with informed consent from each participant. We have inserted the above information in the “Clinical samples and patient information” section of M&M part in page 13-14.

 

Comments from the Reviewer #2

The authors provided a clear and well-written paper. The results and the statistical data provided are scientifically sound and the limitations of their findings were clearly assessed in the discussion section. For this reason, I cannot find major revisions.

However, some minor revisions should be assessed:

1. Page 3 - "Those results confirmed the genetic vibrations in"  variations?

Response: Thank you for your correction, we have corrected the error in the revised version.

2. Page 5 - "potent" is written 3 times in 3 consecutive sentences. Please, use a synonym

Response: Thank you for your suggestion, we have changed “those potent haplotypes” to “those novel haplotypes” in page 5.

3. Page 5 - "Thermo-stability data, the haplotype with rs73005445 can strikingly change their full-length RNA structures (Fig. 3D and 3E)" - Is something missing?

Response: Thank you for your concern, we have revised the paragraph in page 6 to make it clear.

4. Page 5 - Figure 3B is not cited in the text.

Response: Thank you for your concern, 3B is cited in page 6 (Fig. 3A to 3C).

5. Page 7 - Figure 4 labels are very small. It should be better to increase a bit their size.

Response: Thank you for the constructive suggestion. We have increased the label size in Figure 4. We also revised the Fig. 5 and Fig. 6 accordingly to increase the label sizes

6. Page 10 - "Regulation of fatty acid... "  I appreciate a good recap in biochemistry, but I don't understand the point of doing this in the discussion section. Is there some final sentence missing? What's the point of explaining all this and not connecting it with the text? I suggest adding a sentence or rephrasing the entire part.

Response: Thank you for the insightful comment. We removed some biochemistry parts on lipogenesis and focused more on the epigenetic roles of UCA1 variants in endometriosis development as shown in page 12-13.

7. Page 11 - "Blood samples were collected patients"  "Blood samples were collected from patients"

Response: Thank you for your correction, we have corrected the error in the revised version in page 12.

8. Page 12 - "were further enriched by 5"  A square appears after the 5. Some symbol is not recognized, please correct it.

Response: Thank you for founding this error, we have fixed the “µ” sign in the revised version in page 13.

9. Supplementary - I found many small typos in the pathways found: "Giloma" -> "Glioma"?, "Lysince degradation" -> "Lysine degradation"? etc... Please, check carefully the entire document and correct them all.

Response: Thank you for your corrections, we have checked the entire documents and made necessary corrections accordingly.

---

## [Decision Letter · Decision Letter 1]

5 Jul 2022

Genetic variations in UCA1, a lncRNA functioning as a miRNA sponge, determine endometriosis development and the potential associated infertility via regulating lipogenesis

PONE-D-22-10884R1

Dear Dr. Sheu,

We’re pleased to inform you that your manuscript has been judged scientifically suitable for publication and will be formally accepted for publication once it meets all outstanding technical requirements.

Kind regards,

Giovanni Nassa, PhD

Academic Editor

PLOS ONE

Additional Editor Comments (optional):

Reviewers' comments:

Reviewer's Responses to Questions

**Comments to the Author**

1. If the authors have adequately addressed your comments raised in a previous round of review and you feel that this manuscript is now acceptable for publication, you may indicate that here to bypass the “Comments to the Author” section, enter your conflict of interest statement in the “Confidential to Editor” section, and submit your "Accept" recommendation.

Reviewer #1: All comments have been addressed

Reviewer #2: All comments have been addressed

2. Is the manuscript technically sound, and do the data support the conclusions?

Reviewer #1: Yes

Reviewer #2: Yes

3. Has the statistical analysis been performed appropriately and rigorously? 

Reviewer #1: I Don't Know

Reviewer #2: Yes

4. Have the authors made all data underlying the findings in their manuscript fully available?

Reviewer #1: Yes

Reviewer #2: Yes

5. Is the manuscript presented in an intelligible fashion and written in standard English?

Reviewer #1: Yes

Reviewer #2: Yes

6. Review Comments to the Author

Reviewer #1: the authors have adequately addressed my comments raised in a previous round of review and I feel that this manuscript is now acceptable for publication

Reviewer #2: The authors addressed all my comments and I have no other concerns.

I hope they will try to continue on this topic with a bigger cohort.

Good luck

7. PLOS authors have the option to publish the peer review history of their article (what does this mean?). If published, this will include your full peer review and any attached files.

Reviewer #1: No

Reviewer #2: No

---

## [Editor Report · Acceptance letter]

18 Jul 2022

PONE-D-22-10884R1 

Genetic variations in *UCA1*, a lncRNA functioning as a miRNA sponge, determine endometriosis development and the potential associated infertility via regulating lipogenesis 

Dear Dr. Sheu:

I'm pleased to inform you that your manuscript has been deemed suitable for publication in PLOS ONE. Congratulations! Your manuscript is now with our production department. 

Kind regards, 

on behalf of

Dr. Giovanni Nassa 

Academic Editor

PLOS ONE